# NDRG2 Expression in Breast Cancer Cells Downregulates PD-L1 Expression and Restores T Cell Proliferation in Tumor-Coculture

**DOI:** 10.3390/cancers13236112

**Published:** 2021-12-04

**Authors:** Aram Lee, Soyoung Lim, Juyeong Oh, Jihyun Lim, Young Yang, Myeong-Sok Lee, Jong-Seok Lim

**Affiliations:** Department of Biological Science and the Cellular Heterogeneity Research Center, Sookmyung Women’s University, Seoul 04310, Korea; 1215728@sookmyung.ac.kr (A.L.); sornate@sookmyung.ac.kr (S.L.); ryshbwr129@naver.com (J.O.); jhlim@sookmyung.ac.kr (J.L.); yyang@sookmyung.ac.kr (Y.Y.); mslee@sookmyung.ac.kr (M.-S.L.)

**Keywords:** immune checkpoint, PD-L1, NDRG2, breast cancer, T cell proliferation, TCGA data

## Abstract

**Simple Summary:**

N-myc downstream-regulated gene 2 (NDRG2) is a candidate tumor suppressor in various cancers, including breast cancer. Increased expression of programmed death ligand 1 (PD-L1) is frequently observed in human cancers. Despite its role in cancer cells, the effects of NDRG2 on PD-L1 expression and PD-L1-PD-1 pathway disruption have not been investigated. We demonstrated that NDRG2 overexpression inhibits PD-L1 expression in human breast cancer cells. Blocking T cell proliferation by coculture with 4T1 mouse tumor cells that express high levels of PD-L1 could be significantly reversed by NDRG2 overexpression in the same tumor cells. NDRG2 knockdown in NDRG2-transfected cells elicited the upregulation of PD-L1 expression and accelerated the inhibition of T cell proliferation. These findings were confirmed from The Cancer Genome Atlas (TCGA) data that PD-L1 expression in basal and triple-negative breast cancer (TNBC) patients, but not in luminal A or B cancer patients, was negatively correlated with the NDRG2 expression.

**Abstract:**

(1) Background: The aim of the present study was to evaluate the effect of NDRG2 expression in regulating PD-L1 or PD-L2 on malignant breast cancer cells. (2) Methods: Overexpression and knockdown of the NDRG2 gene in human and mouse cancer cells were applied and quantitative real-time PCR and Western blot analysis were performed. T cell proliferation and TCGA analysis were conducted to validate negative correlation of the PD-L1 expression with the NDRG2 expression. (3) Results: We found that NDRG2 overexpression inhibits PD-L1 expression in human breast cancer cells through NF-κB signaling. NDRG2 overexpression in 4T1 mouse breast cancer cells followed by PD-L1 downregulation could block the suppressive activity of cancer cells on T cell proliferation and knockdown of NDRG2 expression enhanced the expression of PD-L1, leading to the inhibition of T cell proliferation by tumor cell coculture. Finally, we confirmed from TCGA data that PD-L1 expression in basal and triple-negative breast cancer patients was negatively correlated with the expression of NDRG2. Intriguingly, linear regression analysis using TNBC cell lines showed that the PD-L1 level was negatively associated with the NDRG2 expression level. (4) Conclusions: Our findings demonstrate that NDRG2 expression is instrumental in suppressing PD-L1 expression and restoring PD-L1-inhibited T cell proliferation activity in TNBC cells.

## 1. Introduction

Programmed cell death protein 1, also known as CD279, is one of the multiple co-inhibitory molecules expressed on the surface of immune-related lymphocytes, such as activated T cells, B cells, and myeloid cells. It binds to two ligands, PD-L1 (CD274) and PD-L2 (CD273), on cancer cells or immune infiltrates [1]. Interestingly, although PD-L2 also contributes to PD-1-mediated T cell inhibition and has a stronger affinity for PD-1 than does PD-L1, antibodies against PD-1, which are able to block binding to both PD-L1 and PD-L2, do not exhibit higher clinical efficacy than antibodies against PD-L1. Thus, in the human tumor microenvironment (TME), PD-L1 is widely believed to be the dominant inhibitory ligand of PD-1 on T cells. Interaction of PD-1 with PD-L1 can affect the activity of T cells in diverse ways, such as by suppressing cytokine production, T cell proliferation, survival, and other effector T cell functions. Therefore, an improved understanding of the mechanisms that regulate PD-L1 expression in tumor cells could lead to better clinical outcomes [2,3]. The effects of checkpoint signaling through PD-1 are reasonably well understood, whereas reverse signaling through PD-L1 within cancer cells has been investigated less than checkpoint signaling through PD-1. Although there are no canonical signaling motifs in cytoplasmic tail of PD-L1, recent literature has implicated the intracellular domain of PD-L1 for evasion of cancer cells from various apoptotic stimuli and its intrinsic signaling properties [4]. For instance, cancer cells with upregulated PD-L1 expression can protect tumor cells from cytotoxic T lymphocyte (CTL)-mediated cytolysis and from the cytotoxic effects of type 1 and type II interferons through reverse PD-L1 signaling within cancer cells, without PD-1 signaling in T cells.

N-myc downstream-regulated gene 2 (NDRG2), a member of the NDRG family, was identified as a stress-responsive gene whose expression is not only downregulated by N-myc gene expression, but also transcriptionally activated by several types of cellular stress stimuli. It has been reported in mammals that there are four types of NDRGs, such as NDRG1, 2, 3, and 4, which all have 57–65% identical amino acid sequences. They have a common NDR domain in the middle of the protein containing an alpha (α)/beta (β) hydrolase-like region [5]. The expression of NDRG2 induced by several stress stimuli, such as hypoxia, DNA damage or endoplasmic reticulum stress (ERS), and other pathological conditions is often associated with its tumor suppressor function in multiple solid tumors, including breast cancer [6], colorectal cancer [7], renal cell carcinoma [8], and lung cancer [9]. It has been also demonstrated that NDRG2 is an important repressor of the PI3K/AKT and NF-κB signaling pathways, which have critical functions in cell proliferation. In particular, our previous study has shown that NDRG2 expression can inhibit breast cancer development by inhibiting tumor cell proliferation, migration, and epithelial-mesenchymal transition (EMT) [10].

Despite the accumulated evidence indicating a tumor suppressive role of NDRG2 in various types of tumors, the question of whether the interaction between tumor cells and immune cells can be influenced by NDRG2 expression in tumor cells remains unanswered. In the present study, we sought to investigate whether NDRG2 expression in aggressive breast tumor cells can influence PD-L1 expression, eventually leading to the alteration of T cell proliferation in response to coculture with tumor cells.

## 2. Materials and Methods

### 2.1. Cell Culture and Reagents

MDA-MB-231 (highly metastatic) and MCF7 (nonmetastatic) human breast cancer cells were obtained from the American Type Culture Collection (ATCC, Manassas, VA, USA). Mouse breast cancer cells (4T1-WT, mock and NDRG2) were kindly provided by Professor KD Kim at Gyeongsang National University in Jinju, Korea. All cell lines were cultured in complete Dulbecco’s modified Eagle’s medium (DMEM; Gibco/Invitrogen, Carlsbad, CA, USA) containing 10% heat-inactivated fetal bovine serum (FBS) (Gibco/Invitrogen) and 1% penicillin/streptomycin (Gibco/Invitrogen). The cells were maintained in an atmosphere of 5% CO_2_ in a 37 °C humidified incubator. PMA (phorbol 12-myristate 13-acetate) and mitomycin C were purchased from Sigma-Aldrich (St. Louis, MO, USA). Anti-mouse PD-L1 (clone 10F.9G2) and IgG2b isotype control antibodies were purchased from BioXcell (West Lebanon, NH, USA).

### 2.2. Overexpression and Knockdown of the NDRG2 Gene in Breast Cancer Cells

MDA-MB-231-NDRG2 cells were previously generated in our laboratory [11]. For the knockdown of NDRG2 by small interfering RNA (siRNA), murine control and NDRG2 siRNAs were purchased from Santa Cruz Biotechnology (Santa Cruz, CA, USA). Control siRNA or NDRG2 siRNA was transfected into 4T1-NDRG2 cells using Lipofectamine™ RNAiMAX (Invitrogen) according to the manufacturer’s recommendations.

### 2.3. Animals

The mouse experiments were conducted according to the guidelines of the Institutional Animal Care and Use Committee after approval by the Institutional Ethical Committee of Sookmyung Women’s University (Resolution No. SMWU-IACUC-1912-024-1). Mouse splenic lymphocytes were obtained from female Balb/c mice at 6 to 8 weeks of age which were purchased from Samtaco (Osan, Korea). The mice were maintained at least for 2 weeks before experiments in SPF facility at Sookmyung Women’s University.

### 2.4. RT-PCR and Quantitative Real-Time PCR

Total RNA was extracted from the harvested cells using TRIzol reagent (Invitrogen) according to the manufacturer’s instructions. For the reverse-transcription reaction, 5 μg of total RNA was converted into first-strand cDNA using M-MLV reverse transcriptase (Promega, Madison, WI, USA), oligo (dT) primers, and dNTPs (Bioneer, Daejeon, Korea). The cDNA was amplified by PCR in a 20 μL reaction mixture containing cDNA, 10 mM dNTPs, 10 pmole of each primer, and 0.5 U of Taq DNA polymerase (Bioneer). All PCR primers were purchased from Bioneer, and the specifically designed PCR primers are shown in Appendix A. The PCR products were electrophoresed on a 1% agarose gel and visualized by ethidium bromide staining.

Quantitative real-time PCR (qRT-PCR) was performed using Power SYBR Green PCR Master Mix according to the manufacturer’s protocol (Applied Biosystems, Foster City, CA, USA). Briefly, the cDNA was diluted with RNase-free water, and Power SYBR Green PCR Master Mix and the indicated primers were added to the sample. The specific primer sequences used for the qRT-PCR are listed in Appendix A. The mixed sample was amplified using the StepOnePlus™ real-time PCR system (Applied Biosystems). GAPDH and cyclophilin were used as endogenous controls.

### 2.5. Western Blot

The protein cell lysates were isolated by Pro-Prep™ reagent (iNtRON Biotechnology, Seongnam, Korea). Total proteins were separated by electrophoresis on gradient SDS-PAGE gels (8% to 15%) and transferred onto a PVDF membrane (Amersham Biosciences, Bukres, UK). Antibodies against GAPDH, NF-κB, NDRG2, p-IKKα/β, Lamin A/C, and α-actinin were purchased from Santa Cruz Biotechnology. Anti-human PD-L1, p-STAT3, STAT3, α-tubulin, p-IκBα, IκBα, p-NF-κB (p-p65), and PD-1 antibodies were purchased from Cell Signaling (Beverly, MA, USA). Anti-mouse PD-L1 antibody was purchased from R&D Systems (Minneapolis, MN, USA). The blots were visualized by Ez-Capture MG (ATTO Corporation, Tokyo, Japan).

### 2.6. Flow Cytometry

The following antibodies were used for flow cytometry analysis. Anti-human PD-L1 (MIH1), anti-human PD-L2 (MIH18), and anti-mouse PD-L1 (MIH5) antibodies were purchased from eBioscience Inc. (San Diego, CA, USA). Cells were incubated with fluorochrome-conjugated mAbs directed against cell surface antigens at 4 °C for 30 min. The cells were washed twice with PBS, and resuspended in FACS buffer. The samples were analyzed using a FACSCanto II flow cytometer (BD Biosciences, San Jose, CA, USA). The flow cytometry data were analyzed by FlowJo software (Tree Star, OR, USA).

### 2.7. CFSE Assay

Red blood cells were removed from the spleen using red blood cell lysis buffer (Sigma-Aldrich). The splenocytes were stained with 2.5 μM CFSE for 7 min at room temperature using a CellTrace^TM^ CFSE Cell Proliferation Kit (Invitrogen). An equal volume of FBS was added to splenocytes and incubated for 3 min. After the cells were washed twice, they were cocultured with 4T1-mock or 4T1-NDRG2 cells in RPMI 1640 (Gibco/Invitrogen) medium supplemented with 10% heat-inactivated FBS. To prevent proliferation of 4T1 cells, cancer cells were pre-treated with mitomycin C for 6 h. CFSE-labeled splenocytes were stimulated using 1 μg/mL anti-CD3 mAb (145-2C11, eBioscience) and 1 μg/mL anti-CD28 mAb (37.51, eBioscience) for 3 days. The cells were stained with an antibody against CD8 (J43, eBioscience), and then CFSE^+^ CD8^+^ T cell proliferation was analyzed by flow cytometry.

### 2.8. Analysis of the Cancer Genome Atlas (TCGA) and Q-Omics Datasets

For the microarray gene expression data, TCGA breast cancer online (https://tcga-data.nci.nih.gov/docs/publications/brca_2012/, 27 February 2012) and “BRCA.exp.547.med.txt.” files were used. Depending on the breast cancer cell types, graph of the expression levels of CD274 and NDRG2 was made. For the linear regression analysis, CD274 and NDRG2 expression data for TNBC or non-TNBC cell lines were obtained from http://qomics.sookmyung.ac.kr, (9 March 2021) [12] and plotted with trend line and equation.

### 2.9. Statistical Analysis

Student’s *t*-test and one-way ANOVA (Tukey’s post-hoc test for multiple comparisons) with GraphPad PRISM software version 9 (GraphPad Software, San Diego, CA, USA) were used for statistical analyses. All the results were obtained from independently repeated experiments and were presented as the mean ± SEM. *p* values of <0.05 were considered statistically significant. 

## 3. Results

### 3.1. NDRG2 Overexpression Inhibits PD-L1 Expression in Human Breast Cancer Cells

Previous reports have shown that PD-L1 expression is increased in triple-negative breast cancer (TNBC) patients and TNBC cell lines [13]. To elucidate the relationship between NDRG2 expression and PD-L1 activity in malignant breast cancer cells, MDA-MB-231 cells, a TNBC cell line, were stably transfected with an empty vector or the NDRG2 vector. The expression levels of PD-L1, PD-L2, PD-1, and NDRG2 were analyzed by RT-PCR or real-time PCR. As shown in Figure 1A–D, the mRNA expression levels of PD-L1 and PD-1 were significantly downregulated in the NDRG2-transfected cells, whereas PD-L2 expression was not affected by NDRG2 overexpression. Similarly, the protein expression of PD-L1 was markedly reduced in NDRG2-overexpressing cells and NDRG2-positive MCF7 cells, which showed low expression of pSTAT3 (Figure 1E). To verify the effects of NDRG2 overexpression in breast cancer cells, we measured the expression of PD-L1 and PD-L2 by flow cytometry. PD-L1 and PD-L2 expression levels on the cell surface were significantly decreased in NDRG2-ovexpressing cells (Figure 1F,G). Consistent with the RT-PCR experiment in Figure 1, although PD-L2 was expressed on MDA-MB-231 cells, its expression level was quite low compared with the level of PD-L1 expression. These findings strongly support the hypothesis that NDRG2 negatively regulates the expression of PD-L1 in malignant breast cancer cells.

### 3.2. NDRG2 Downregulates PD-L1 Expression through NF-κB Signaling

Our previous study demonstrated that NDRG2 inhibits NF-κB signaling in breast cancer cells [11]. In addition, previous reports have shown that PD-L1 expression is regulated through the STAT3 or NF-κB signaling pathway in various cancer cells, such as non-small lung cancer [14,15], melanoma [16], and breast cancer [17]. Based on this, we investigated whether NF-κB signaling is involved in NDRG2-mediated PD-L1 downregulation. First, we confirmed the expression level of NF-κB signaling-related genes in MDA-MB-231-NDRG2 and mock cells. In line with the low expression of PD-L1, NDRG2-ovexpressing cells exhibited low expression of p-IκBα, p-IKKα/β, and p-p65 compared to mock cells (Figure 2A). In addition, p-p65 was also decreased in nuclear compartments in NDRG2-ovexpressing cells, suggesting that NF-κB signaling is indeed suppressed by NDRG2 overexpression (Figure 2B). Next, we utilized phorbol 12-myristate 13-acetate (PMA) to induce NF-κB signaling. Treatment of mock cells with PMA induced phosphorylation of IκBα and IKKα/β, whereas it could not elicit NF-κB signaling in NDRG2-overexpressing cells (Figure 2C). Notably, PMA treatment increased PD-L1 expression in MDA-MB-231 cells. The PD-L1 expression level peaked at 6–12 h and then decreased at 24 h. However, PMA treatment did not induce PD-L1 expression in NDRG2-ovexpressing cells (Figure 2D). Moreover, PMA-induced PD-L1 expression on the surface of MDA-MB-231 cells was strongly decreased by NDRG2 overexpression (Figure 2E). Collectively, these data demonstrate that NDRG2 downregulates PD-L1 expression by suppressing the NF-κB signaling pathway.

### 3.3. PD-L1 Expression Is Suppressed by NDRG2 Overexpression in Mouse 4T1 Mammary Tumor Cells

To confirm whether the inhibitory effect of NDRG2 overexpression on PD-L1 expression was related to the regulation of STAT3 or NF-κB activation in mouse breast cancer cells, we examined the phosphorylation levels of STAT3 and NF-κB in 4T1-mock and -NDRG2 cells. As shown in Figure 3A,B, NDRG2 overexpression in 4T1 cells downregulated the activation of STAT3 and NF-κB as well as PD-L1 expression. Collectively, these results unambiguously show that PD-L1 and active forms of STAT3 and NF-κB are strongly expressed in malignant breast cancer cells, but their expression is significantly reduced by NDRG2 overexpression.

### 3.4. T Cell Proliferation Reduced by Tumor Cell Coculture after anti-CD3/CD28 Stimulation Is Notably Recovered by NDRG2 Overexpression in Tumor Cells

To evaluate in vitro anti-tumor effects based on the anti-PD-L1 mAb in the 4T1 mouse mammary carcinoma model, we measured the proliferation of CD8^+^ T cells after anti-CD3/CD28 antibody stimulation. As expected, T cell proliferation was dramatically suppressed in 4T1 coculture and restored in 4T1 coculture treated with anti-PD-L1 antibody in a dose-dependent manner (Figure 4A), suggesting that 4T1 cells have a suppressive activity in the T cell proliferation dependent on PD-1/PD-L1 binding. Next, we assessed the effect of NDRG2 overexpression in breast cancer cells on T cell proliferation. To investigate whether NDRG2 overexpression in 4T1 cells is able to induce the proliferation of T cells, splenocytes of Balb/c mice were prelabeled with CFSE and then cocultured with 4T1-mock or -NDRG2 cells. As shown in Figure 4B,C, 4T1-mock markedly suppressed the proliferation of CD8^+^ T cells in response to stimulation with anti-CD3 and anti-CD28 antibodies, and this effect was most obvious after 2 or 3 days of incubation with 4T1 cells. In contrast, NDRG2 overexpression in 4T1 cells decreased the suppressive effect of cancer cells on CD8^+^ T cell proliferation (with a 4T1 cell to splenocyte ratio of 1:10) (Figure 4D). These findings indicate that NDRG2 overexpression in 4T1 cells followed by PD-L1 downregulation could block the suppressive activity of cancer cells on T cell proliferation and rescue the proliferation activity of T cells by stimulation with antibodies.

### 3.5. Knockdown of NDRG2 Expression in Tumor Cells Enhances the Expression Level of PD-L1, Leading to the Inhibition of T cell Proliferation by Tumor Cell Coculture

Because NDRG2 overexpression affected PD-L1 expression in breast cancer cells, we tested whether NDRG2 downregulation was able to influence PD-L1 expression on tumor cells. The inhibition of NDRG2 expression by NDRG2 siRNA increased phosphorylation of NF-κB protein expression level in 4T1 cells (Figure 5A). Intriguingly, the inhibition of NDRG2 expression in 4T1-NDRG2 cells by NDRG2 siRNA induced an increase in both the protein level and the surface expression of PD-L1 (Figure 5A,B). To determine whether NDRG2 knockdown affects the ability of cancer cells to suppress CD8^+^ T cell proliferation, splenocytes treated with anti-CD3 and anti-CD28 antibodies were cocultured with transfected 4T1 cells. As expected, CD8^+^ T cell proliferation was significantly reduced by NDRG inhibition compared with T cell proliferation induced by control tumor cells (with a 4T1 to splenocyte ratio of 1:5) (Figure 5C,D). Therefore, NDRG2 downregulation seems to decrease CD8^+^ T cell proliferation by restoring PD-L1 expression. Collectively, our findings demonstrated that NDRG2 might have a crucial role in preventing the suppressive activity of cancer cells on T cell proliferation by modulating PD-L1 expression.

### 3.6. PD-L1 Expression in Basal or Triple-Negative Breast Cancer Tissues Is Inversely Correlated with NDRG2 Expression Levels

We next investigated The Cancer Genome Atlas (TCGA) datasets to find the correlations among NDRG2, PD-L1, and PD-L2 mRNA expression in human breast cancer patients. Dataset analysis revealed that PD-L2 expression in basal and triple-negative breast cancer (TNBC) patients, but not in luminal A or B patients, showed a very weak negative correlation with the expression of NDRG2, suggesting a minor role of PD-L2 expression in the progression of breast tumors (data not shown). However, consistent with the previous results, there was a significant negative correlation between the expressions of NDRG2 and PD-L1. Of note, the basal and TNBC subtypes exhibited relatively strong negative correlations (PCC value −0.607 to −0.319) compared to the luminal A (−0.046), luminal B (−0.092), and HER2 (−0.334) subtypes (Figure 6A–F). To further investigate the correlation between these two genes in TNBC, we performed linear regression analysis using 28 human TNBC cell lines, including MDA-MB-231, −436, −468, BT549, BT20, HCC1143, and CAL120. Notably, the PD-L1 level was negatively associated with the NDRG2 expression level in all of these cell lines (PCC = −0.46) (Figure 6G, left). However, there was no significance between PD-L1 and NDRG2 expression in non-TNBC cell lines (PCC = 0.03) (Figure 6G, right). These data show that NDRG2 expression is inversely correlated with PD-L1 expression, especially in TNBC. Collectively, these findings demonstrate that NDRG2 expression is instrumental in suppressing PD-L1 expression and restoring PD-L1-inhibited T cell proliferation activity in TNBC cells.

## 4. Discussion

Programmed death ligand 1 (PD-L1), a critical immune checkpoint protein, binds to programmed death 1 (PD-1) on T cells, leading to cancer immunosuppression [18]. Interactions between the extracellular domains of PD-L1 and PD-1 can induce a conformational change in PD-1, leading to phosphorylation of the cytoplasmic immunoreceptor tyrosine-based inhibitory motif (ITIM) and the immunoreceptor tyrosine-based switch motif (ITSM) by Src family kinases [19]. These phosphorylated tyrosine motifs subsequently recruit the SHP-2 and SHP-1 protein tyrosine phosphatases to attenuate T cell-activating signals. High expression of PD-L1 protein levels that promote the immune escape of cancer cells is usually observed in different types of cancers. In particular, PD-L1 expression in cancer cells is regulated by multiple signaling pathways, including NF-κB, MAPK, mTOR, STAT, and cMyc [2,20]. In the present study, we demonstrated that PD-L1 expression on breast cancer cells was significantly downregulated by the overexpression of NDRG2, leading to the prevention of tumor cell-induced interference of splenic T cell proliferation stimulated with a combination of anti-CD3/CD28 antibodies. Multiple reports have already demonstrated that NDRG2 expression in cancer cells is able to inhibit the NF-κB, MAPK, and STAT signaling pathways [11,21,22], thus raising the hypothesis that NDRG2 expression may hamper PD-L1 expression in tumor cells. In line with this hypothesis, PD-L1 expression in MBA-MB-231 breast cancer cells was significantly influenced by the modulation of NDRG2 expression, and it had a remarkable impact on the proliferation activity of stimulated splenic T cells after coculturing with mouse breast tumor cells. In addition to T cell proliferation, a significant negative association between NDRG2 and PD-L1 expression was observed in basal and triple-negative types of human breast cancer tissues through TCGA database analysis. Furthermore, we found from linear regression analysis using 28 human TNBC cell lines that the PD-L1 level was negatively associated with the NDRG2 expression level. Therefore, our findings clearly indicate that the tumor suppressive role of NDRG2 may be in part due to its inhibitory effect on PD-L1 expression in tumor cells, and that it is intimately involved in directly controlling T cell activity in the tumor microenvironment.

Increasing evidence has shown that the PD-L1 protein undergoes degradation in proteasomes or lysosomes by multiple pathways, leading to strongly increased effectiveness of cancer immunotherapy [23]. Although it is well known that PD-L1 is also subjected to ubiquitination and degradation, cancer cells often exhibit the ability to inhibit this process. For instance, in the tumor microenvironment, TNF-α secreted by macrophages activates NF-κB in cancer cells, leading to increased deubiquitinase CSN5 gene transcription and expression [24]. It stabilizes the PD-L1 protein by inhibiting its ubiquitination and degradation, resulting in cancer cell immune escape. In addition, PD-L1 glycosylation also plays an important role in stabilizing the PD-L1 protein, which subsequently inhibits PD-L1 degradation in cancer stem cells, resulting in reduced T cell activity [25]. Interestingly, NDRG2 in colorectal cancer could affect the complex components of ubiquitin-protein ligase E3A (UBE3A) and estrogen receptor beta (ERβ), reducing the ubiquitin-mediated proteasome degradation of ERβ, demonstrating that NDRG2 could bind to UBE3A to hinder the binding of UBE3A to ERβ [26]. In contrast to the inhibitory effect of NDRG2 on ubiquitin-protein ligase, a recent report demonstrated in colitis and colitis-associated tumors that NDRG2 enhances the interaction of the E3 ligase FBXO11 with Snail, the repressor of E-cadherin, to promote Snail degradation by ubiquitination and to maintain E-cadherin expression [27]. Thus, it would be interesting to address the question of whether NDRG2 has a crucial role in regulating PD-L1 degradation in proteasomes or lysosomes. In addition to proteasome-dependent degradation, autophagic degradation of PD-L1 associated with lysosome biogenesis plays a role in protein stability [28]. Whether the downregulation of PD-L1 is related to increased autophagy activity by NDRG2 remains to be explored in future studies, even though decreased NDRG2 expression in prostate cancer cells revealed an increase in autophagy and cell viability and a decrease in cell apoptosis [29].

Finally, since multiple soluble mediators, such as TNF-α, IFN-γ, IL-1β, IL-17, and IL-27, induce PD-L1 expression on tumor cells [30,31], it may be reasonable to raise the question of whether the expression of certain mediators among them is downregulated by NDRG2 expression. However, there has been no clear evidence indicating such a role of specific cytokines. In a previous report, we demonstrated that NDRG2 expression in histiocytic leukemia cells could negatively regulate the IL-10 signaling pathway, which is modulated via SOCS3 and STAT3 [32]. Indeed, it has been reported that although IL-10 itself does not directly induce the expression of PD-L1, after inhibiting the expression of IL-10 in the environment, the expression level of PD-L1 also decreases, suggesting that IL-10 can induce the expression of PD-L1 via certain factors present in cells [33,34]. Therefore, it is reasonable to assume that the NDRG2-induced reduction in IL-10 signaling may be associated with PD-L1 downregulation in breast cancer cells. In line with this assumption, we observed increased expression of IL-10 and PD-L1 in peritoneal macrophages followed by treatment with tumor-conditioned culture medium (TCCM) obtained from mock-transfected 4T1 breast cancer cells, whereas an increase in IL-10 and PD-L1 expression was not observed in those treated with TCCM from NDRG2-transfected cells (data not shown).

In conclusion, we investigated whether NDRG2 expression in aggressive breast tumor cells could influence PD-L1 expression, eventually leading to an alteration of T cell proliferation in response to coculture with tumor cells. NDRG2 expression in breast cancer could reduce PD-L1 expression and restore the T cell proliferation activity suppressed by PD-L1 expression on tumor cells, indicating a possible role of NDRG2 in PD-L1 downregulation. TCGA data analysis of basal and triple-negative types of human breast cancers revealed a significant negative correlation between NDRG2 and PD-L1 expression, which was, again, confirmed by regression analysis using multiple human TNBC cell lines. Collectively, our data support the hypothesis that NDRG2 plays a crucial role as a tumor suppressor via decreased expression of immune checkpoint molecules that are involved in antitumor immune responses.

## 5. Conclusions

In the present study, it was confirmed that downregulation of NDRG2, which is often observed in several types of cancer patients, is associated with upregulation of PD-L1 expression in breast cancer cells and patients. NDRG2 expression in breast cancer cells could restore the T cell proliferation activity that is suppressed by PD-L1 expression on tumor cells.

## Figures and Tables

**Figure 1 cancers-13-06112-f001:**
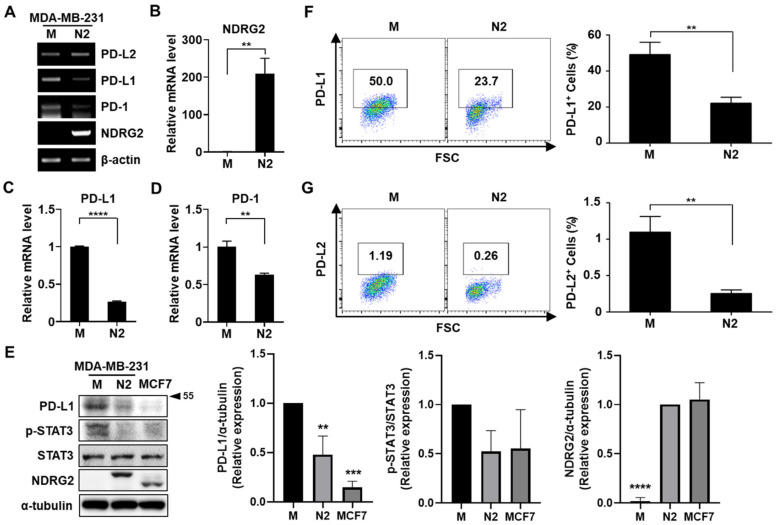
NDRG2 overexpression inhibits PD-L1 expression in human breast cancer cells. (**A**–**D**) The mRNA expression of NDRG2, PD-L1, and PD-1 in MDA-MB-231-mock (M) and MDA-MB-231-NDRG2 (N2) cells was examined by RT-PCR (**A**) and real-time PCR (**B**–**D**). (**E**) The protein expression of NDRG2, PD-L1, and p-STAT3 in MDA-MB-231-mock, -NDRG2, and MCF7 cells was measured by Western blot analysis. (**F**,**G**) MDA-MB-231-mock and MDA-MB-231-NDRG2 cells were stained with specific antibodies against PD-L1 or PD-L2 and analyzed by flow cytometry. ****
*p* < 0.01, *** *p* < 0.001, and ******
*p* < 0.0001. The uncropped blots and molecular weight markers are shown in Appendix A.

**Figure 2 cancers-13-06112-f002:**
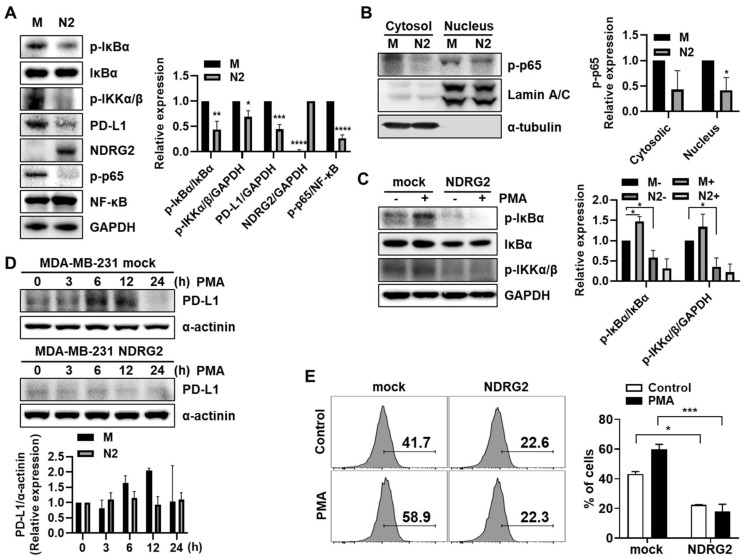
The inhibitory effects of NDRG2 on PD-L1 expression through NF-κB signaling**.** (**A**) Western blotting was performed with MDA-MB-231 mock or NDRG2 whole cell lysates to measure p-IκBα, IκBα, NF-κB, p-IKKα/β, p-p65, PD-L1, and NDRG2 expression. (**B**) The cell lysates were fractionated into nuclear and cytosolic compartments to assess p-p65 expression levels. (**C**) Cells were treated with 40 ng/mL of PMA for 1 h to measure the p-IκBα and p-IKKα/β expression levels. (**D**) PMA was administered for the indicated times to evaluate the effect of PMA on PD-L1 expression, and Western blotting was performed with whole-cell lysates. (**E**) After 12 h of PMA stimulation, the cells were harvested, and the PD-L1 expression on the surface of MDA-MB-231 cells was measured by flow cytometry. All experiments were independently repeated at least 3 times. * *p* < 0.05, ****
*p* < 0.01, *** *p* < 0.001, and **** *p* < 0.0001. The uncropped blots and molecular weight markers are shown in Appendix A.

**Figure 3 cancers-13-06112-f003:**
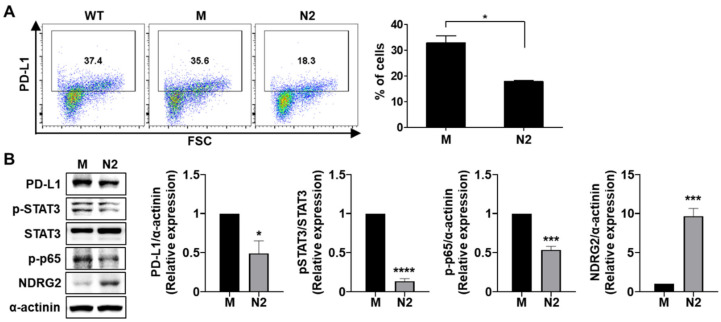
NDRG2 overexpression reduces the expression of PD-L1, p-STAT3, and p-p65. (**A**) PD-L1 expression in 4T1-wild-type, -mock (M), and -NDRG2 (N2) cells was measured by flow cytometry. (**B**) The expression levels of NDRG2, p-STAT3, and p-p65 (p-NF-κB) in 4T1-mock and 4T1-NDRG2 cells were examined by Western blot analysis. All experiments were independently repeated at least 3 times. ** p* < 0.05, *** *p* < 0.001, and **** *p* < 0.0001. The uncropped blots and molecular weight markers are shown in Appendix A.

**Figure 4 cancers-13-06112-f004:**
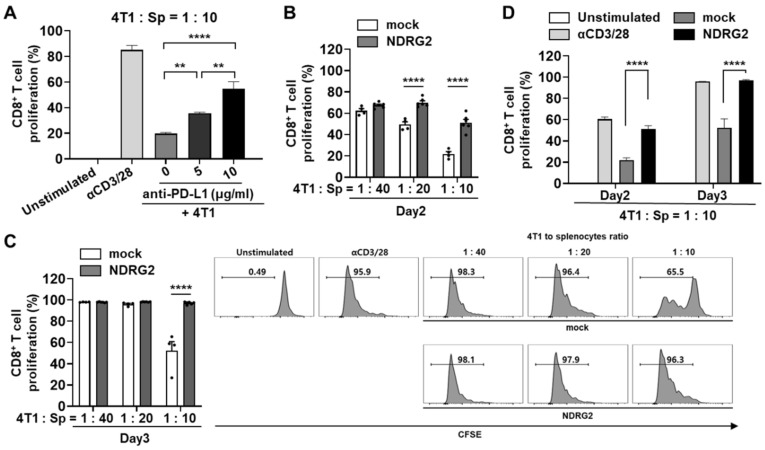
NDRG2 overexpression in breast cancer cells is involved in T cell proliferation. (**A**–**D**) Splenocytes were obtained from Balb/c mice and the RBCs were removed. (**A**) After 24 h of exposure to anti-PD-L1 mAb, the 4T1 cells were cocultured with CFSE-labeled splenocytes at 1:10 ratio for 3 days. The cells were harvested and analyzed by flow cytometry. (**B**–**D**) The splenocytes were labeled with CFSE and cocultured with 4T1 cells for 2 or 3 days. The cells were harvested and analyzed by flow cytometry. The cells were gated on CD8^+^ cells, and the histogram represents the CFSE level. All experiments were independently repeated at least 3 times. ** *p* < 0.01 and **** *p* < 0.0001.

**Figure 5 cancers-13-06112-f005:**
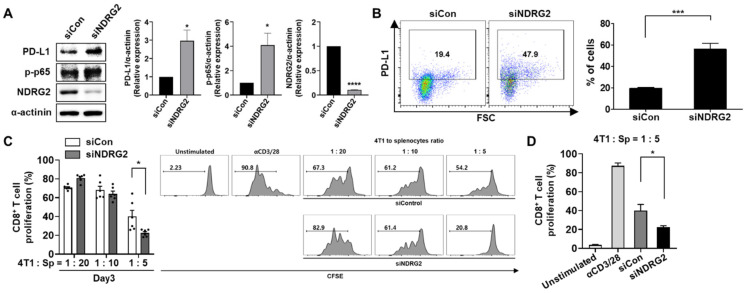
siRNA-mediated knockdown of NDRG2 restores PD-L1 expression and T cell suppression activity. 4T1-NDRG2 cells transiently transfected with control or NDRG2 siRNA were cultured in fresh medium (DMEM or RPMI) supplemented with 10% FBS. (**A**) Protein levels of NDRG2, p-p65, and PD-L1 in 4T1-siControl and 4T1-NDRG2–siRNA transfectants were measured by Western blot analysis. (**B**) PD-L1 expression on the surface was confirmed by flow cytometry. (**C**,**D**) Splenocytes were stained with CFSE and cocultured with transfected 4T1 cells for 3 days. The cells were harvested and analyzed by flow cytometry. The cells were gated on CD8^+^ cells, and the histogram represents the CFSE level. All experiments were independently repeated at least 3 times. ** p* < 0.05, **** p* < 0.001, and ***** p* < 0.0001. The uncropped blots and molecular weight markers are shown in Appendix A.

**Figure 6 cancers-13-06112-f006:**
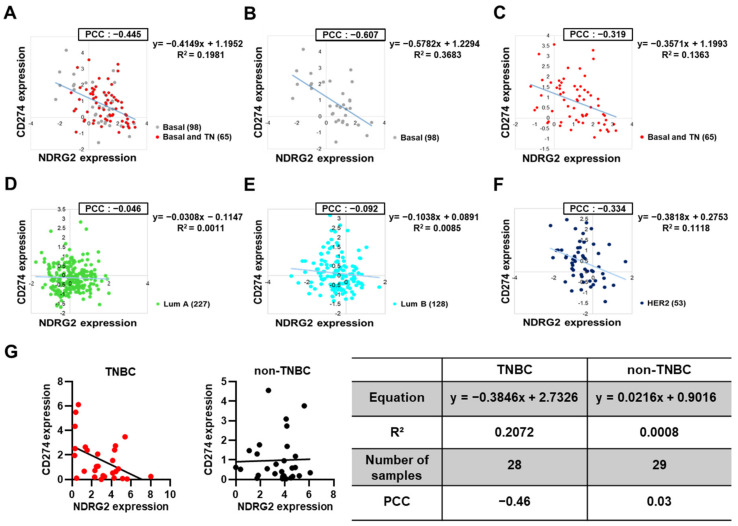
Negative correlation of NDRG2 and PD-L1 expression in human breast cancer patients. (**A**–**F**) Scatter plot showing a negative correlation between NDRG2 and PD-L1 expression in breast cancer patients from the TCGA dataset analysis. Pearson correlation coefficient (PCC) values between two genes and the related trendlines are depicted with equations. (**G**) Linear regression analysis of NDRG2 and PD-L1 expression levels in TNBC or non-TNBC cell lines. Number of samples, trendline equations, R^2^ (coefficient of determination), and PCC values are shown in a separate table.

## Data Availability

The data presented in this study are available on request from the corresponding author.

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
