# Peer review of "NDRG2 Expression in Breast Cancer Cells Downregulates PD-L1 Expression and Restores T Cell Proliferation in Tumor-Coculture"

_cancers, 2021, doi:10.3390/cancers13236112_

Round 1

Reviewer 1 Report

The manuscript was revised according to the reviewers' suggestions and the authors responded well to all the reviewers' comments.

Reviewer 2 Report

Authors have addressed all comments

This manuscript is a resubmission of an earlier submission. The following is a list of the peer review reports and author responses from that submission.

Round 1

Reviewer 1 Report

In this manuscript, the authors show that N-myc downstream-regulated gene 2 (NDRG2) regulates PD-L1 expression via STAT3 and NF-κB signaling pathways in triple negative (TN) breast cancer cells by in vitro experiments. Furthermore, the interaction between NDRG2 and PD-L1 was confirmed in The Cancer Genome Atlas (TCGA) data of TN breast cancer. Although the interaction between NDRG2 and PD-L1 expression in breast cancer cells is interesting, the biological and functional role of NDRG2 in immune function needs to be better defined. Comments are as follows.

  1. In vitro experiments using human and mouse breast tumor cells clearly demonstrated that NDRG2 regulates the expression of PD-L1 in breast tumor cells, as the experimental methods were properly designed and performed. However, the biological function of NDRG2 in tumor progression and metastasis of TN breast cancer is unknown.
  2. The interaction between NDRG2 and PD-L1 expression showed an inverted correlation: how is NDRG2 expression regulated in breast cancer cells? In the immunosuppressed state of the tumor microenvironment, the expression of NDRG2 is reduced and PD-L1 is expressed, resulting in immune escape of tumor cells.
  3. Indeed, expression of PD-L1 in tumor cells inhibits the cytotoxic function of effector T cells, but can downregulation of PD-L1 by overexpression of NDRG2 restore the immune function of T cells in breast cancer patients?
  4. In Figure 2A and Figure 2B, the blot analysis of p-p65 is obscured due to the dark background and needs to be replaced with a clearer figure. Similarly, in Figure 2D, the blot analysis of PD-L1 is also unclear. The size of the protein needs to be indicated in the figure.
  5. The clinical significance of NDRG2 in the treatment of breast cancer is not clear; could targeting NDRG2 improve the therapeutic efficacy of breast cancer, especially TN breast cancer?
  6. There are several misspellings in the text (e.g., P3, L74).
  7. In order to evaluate the functional role of NDRG2 in the treatment of breast cancer, it would be interesting to see what happened to NDRG2 expression after treatment with anticancer agents or immune checkpoint inhibitors.

Author Response

Review 1

Comments and Suggestions for Authors

In this manuscript, the authors show that N-myc downstream-regulated gene 2 (NDRG2) regulates PD-L1 expression via STAT3 and NF-κB signaling pathways in triple negative (TN) breast cancer cells by in vitro experiments. Furthermore, the interaction between NDRG2 and PD-L1 was confirmed in The Cancer Genome Atlas (TCGA) data of TN breast cancer. Although the interaction between NDRG2 and PD-L1 expression in breast cancer cells is interesting, the biological and functional role of NDRG2 in immune function needs to be better defined. Comments are as follows.

1. In vitro experiments using human and mouse breast tumor cells clearly demonstrated that NDRG2 regulates the expression of PD-L1 in breast tumor cells, as the experimental methods were properly designed and performed. However, the biological function of NDRG2 in tumor progression and metastasis of TN breast cancer is unknown.

- We sincerely thank you for the comment. As described in the manuscript, NDRG2 is a tumor suppressor gene that suppresses progression and metastasis of triple negative breast cancer. Kim et al. demonstrated that NDRG2 inhibits the promoter activation of Snail through STAT3 signaling, which results in attenuation of migration and invasion of triple negative breast cancer cells (N-myc downstream-regulated gene 2 (NDRG2) suppresses the epithelial–mesenchymal transition (EMT) in breast cancer cells via STAT3/Snail signaling, Cancer Letters, 354: 33-42, 2014). Furthermore, increasing numbers of review articles on NDRG2 function demonstrate a significant role of NDRG2 in diverse cancers (Cells 10(10):2649, 2021; Histol Histopathol. 33(7):655-663, 2018; Apoptosis 21(6):675-82, 2016).

2. The interaction between NDRG2 and PD-L1 expression showed an inverted correlation: how is NDRG2 expression regulated in breast cancer cells? In the immunosuppressed state of the tumor microenvironment, the expression of NDRG2 is reduced and PD-L1 is expressed, resulting in immune escape of tumor cells.

- Thank you for the comment. Previous study reported that NDRG2 expression is downregulated in several breast cancer cells by hypermethylation and mutation in the promoter region of NDRG2 (Promoter methylation, mutation, and genomic deletion are involved in the decreased NDRG2 expression levels in several cancer cell lines. Biochem Biophys Res Commun. 358(1):164-169, 2007). In addition, since STAT3 that is suppressed by NDRG2 expression also plays a role in the transcriptional regulation of PD-L1 expression, we checked STAT3 activation in relation to PD-L1 expression in the current study.

3. Indeed, expression of PD-L1 in tumor cells inhibits the cytotoxic function of effector T cells, but can downregulation of PD-L1 by overexpression of NDRG2 restore the immune function of T cells in breast cancer patients?

- As the reviewer suggested, we interrogated immune cell infiltration in breast cancer patients using Tumor Immune Estimation Resource (TIMER; cistrome.shinyapps.io/ timer). However, we failed to gain meaningful difference of T cell infiltration between NDRG2-high and -low groups. We think the question of whether in vivo overexpression of NDRG2 can affect the immune function of T cells in breast cancer patients needs to be further illuminated in future research.

 4. In Figure 2A and Figure 2B, the blot analysis of p-p65 is obscured due to the dark background and needs to be replaced with a clearer figure. Similarly, in Figure 2D, the blot analysis of PD-L1 is also unclear. The size of the protein needs to be indicated in the figure.

- Thank you for the suggestion. In response to the reviewer’s suggestion, we tried to make clearer western blots and partially performed new experiments to produce clear figures. More importantly, we added new figures showing a quantitative comparison of band intensity to provide relative expression in Figure 1E, Figure 2A, B, C, D, Figure 3B, and Figure 5A. The size of each protein band was indicated in original gel data submitted in a separate file.

5. The clinical significance of NDRG2 in the treatment of breast cancer is not clear; could targeting NDRG2 improve the therapeutic efficacy of breast cancer, especially TN breast cancer?

- Thank you for the comment. Several studies implicated the importance of NDRG2 expression in therapeutic efficacy of various cancers, including breast cancer. For example, in breast cancer, NDRG2 improves chemo-resistance by upregulating Bad expression (Reference: “NDRG2 promotes adriamycin sensitivity through a Bad/p53 complex at the mitochondria in breast cancer.” Oncotarget 8(17), 29038-29047, 2017). In addition, we found in the present study that NDRG2 gene expression is negatively associated with CD274 gene expression in TNBC, but not in non-TNBC, suggesting that the regulation of NDRG2 expression could be a novel therapeutic strategy to prevent immune escape of TNBC tumor cells. Accordingly, we added data and changed the Figure 6G as follows.

6. There are several misspellings in the text (e.g., P3, L74).

- In response to the reviewer’s comment, we corrected several misspellings in the text. We are very grateful to the reviewer for helpful comments.

7. In order to evaluate the functional role of NDRG2 in the treatment of breast cancer, it would be interesting to see what happened to NDRG2 expression after treatment with anticancer agents or immune checkpoint inhibitors.

-Thank you for your suggestion. Given the above mentioned reference showing anticancer drug sensitivities, it would be very interesting and we will further investigate in the future study.

Reviewer 2 Report

Lee et al have demonstrated that NDRG2 inhibits PD-L1 expression and restore the  T cell proliferation in the presence of TNBC cells. Findings are interesting and experiments are well designed. However, there are major issues that need to be addressed.

1) Figure legends are not clear enough. There is no indication of what is M and N2 throughout the manuscript. This must be indicated at least in the figure legend and preferably also in the results /methods section, which indicated that clone 16 has the best level of NDRG2 overexpression. The relationship between clone 16 and N2 is not clear. In addition is should be indicated how many times the experiments are done.

2) Western blots need to be substantially improved. Usually, experiments are done at least three times to ensure that results are consistent. It is hard to believe that this is the best representative figure the authors have. In Figure 1E, it is not clear why NDRG2 has a higher molecular weight in N2 as compared with endogenous NDRG2 in MCF7 and MDA-MB-231. In Figure 2A, two bands are shown for PD-L1 in western blot, which is different from the one band presented in Figure 1E, although, in both situations, the protein comes from MDA-MB-231 cells. Similarly, multiple bands are shown for p-IKKa/b where part of them are cropped, making it confusing. In the same Figure p-p65 is hardly seen due to low signal and high noise/background. Again, two bands are shown for PD-L1 in the western blot presented in Figure 2D and this time the lower band is more prominent than the upper one. Surprisingly, the PD-L1 band can hardly be seen in untreated cells, contrary to the stable and clear PD-L1 expression shown in Figures 1E and 2A. In addition, the noise/background in most of the blots is very high. Figure 2E, the y axis legend is not clear (is it PDL1 expression??)

3) In figures 4 and 5 an additional control must be added: the treatment with PD-L1 antibody to block the interaction of PD-L1 with PD-1. This control is essential as NDRG2 can affect the proliferation rate of cancer cells (4T1), leading to consumption of the nutrients and production of waste products which could inhibit proliferation of lymphocytes independent of PD-L1/PD-1 interaction.

4) it is not clear why there is no consistence in the used housekeeping gene as sometimes it is Beta actin, then alpha tubulin then GAPDH then alpha actinin

5) There is contraction in relation to PD-L2 expression in the presence of absence of NDRG2 as in Figure 1A it seems no difference while in 1F clearly there is decrease to the same extent as PD-L1 although the initial level of expression is different.  PD-L2 data seems not convincing, irrelevant and preferably removed from the manuscript.

Author Response

Review 2

Comments and Suggestions for Authors

Lee et al have demonstrated that NDRG2 inhibits PD-L1 expression and restore the T cell proliferation in the presence of TNBC cells. Findings are interesting and experiments are well designed. However, there are major issues that need to be addressed.

1) Figure legends are not clear enough. There is no indication of what is M and N2 throughout the manuscript. This must be indicated at least in the figure legend and preferably also in the results /methods section, which indicated that clone 16 has the best level of NDRG2 overexpression. The relationship between clone 16 and N2 is not clear. In addition is should be indicated how many times the experiments are done.

- We sincerely thank you for the comments. In response to the reviewer’s comment, we deleted the sentence in “Material and Methods” section to avoid misunderstanding. In addition, we added the following sentences in the legend for Fig. 2, 3, 4, and 5 indicating results derived from repeated experiments.

All experiments were independently repeated at least 3 times.

2) Western blots need to be substantially improved. Usually, experiments are done at least three times to ensure that results are consistent. It is hard to believe that this is the best representative figure the authors have. In Figure 1E, it is not clear why NDRG2 has a higher molecular weight in N2 as compared with endogenous NDRG2 in MCF7 and MDA-MB-231. In Figure 2A, two bands are shown for PD-L1 in western blot, which is different from the one band presented in Figure 1E, although, in both situations, the protein comes from MDA-MB-231 cells. Similarly, multiple bands are shown for p-IKKa/b where part of them are cropped, making it confusing. In the same Figure p-p65 is hardly seen due to low signal and high noise/background. Again, two bands are shown for PD-L1 in the western blot presented in Figure 2D and this time the lower band is more prominent than the upper one. Surprisingly, the PD-L1 band can hardly be seen in untreated cells, contrary to the stable and clear PD-L1 expression shown in Figures 1E and 2A. In addition, the noise/background in most of the blots is very high. Figure 2E, the y axis legend is not clear (is it PDL1 expression??)

- Thank you for the comment and suggestion. In response to the reviewer’s suggestion, we tried to make clearer western blots and partially performed new experiments to produce clear figures. More importantly, we added new figures showing a quantitative comparison of band intensity to provide relative expression in Figure 1E, Figure 2A, B, C, D, Figure 3B, and Figure 5A. The size of each protein band was indicated in the original gel data submitted in a separate file. Regarding molecular weight of NDRG2 protein, we transfected MDA-MB-231 cells with the plasmid pCMV-Taq2B/human NDRG2, thus leading to the expression of higher molecular weight protein compared to endogenous NDRG2 in MCF7. With respect to PD-L1 identification, there were in fact a few bands on western blot when using anti-PD-L1 antibody. As seen in the original gel, more than two bands are mainly presented in western blot as reported in other literatures. Since the change of band intensity was usually consistent in most experiments, we chose only the upper band in the revised manuscript.

 - Regarding p-p65 and p-IKKa/b in Fig. 2B and 2C, we tried to minimize the background noise in the revision. Nonetheless, we think that the change of band intensities was consistent and well worth enough to mention the NDRG2 transfection effect.

 - Regarding the label on Y-axis in Figure 2E, Y-axis label should be % of PD-L1-positive cells. Accordingly, new label was inserted into Y-axis in Fig. 2E.

3) In figures 4 and 5 an additional control must be added: the treatment with PD-L1 antibody to block the interaction of PD-L1 with PD-1. This control is essential as NDRG2 can affect the proliferation rate of cancer cells (4T1), leading to consumption of the nutrients and production of waste products which could inhibit proliferation of lymphocytes independent of PD-L1/PD-1 interaction.

- We thank you for the comments. Unfortunately, we forgot to describe the mitomycin C pretreatment of tumor cells to avoid consumption of the nutrients and production of waste products which could inhibit proliferation of splenic lymphocytes independent of PD-L1/PD-1 interaction. Thus, we added the following sentence in CSFE assay of “Material and Methods” section.

After the cells were washed twice, they were cocultured with 4T1-mock or 4T1-NDRG2 cells in RPMI 1640 (Gibco/Invitrogen) medium supplemented with 10% heat-inactivated FBS. To prevent proliferation of 4T1 cells, cancer cells were pre-treated with mitomycin C for 6 h.

- In addition, as suggested by the reviewer, we performed the inhibition assay using anti-PD-L1 blocking antibody to confirm the PD-L1-mediated inhibition of T cell proliferation. As expected, we found that the proliferation was rescued by a treatment with anti-PD-L1 antibody in a dose-dependent manner. Accordingly, we added new figure in Figure 4A and inserted the sentences in “Results” section and the legend for Figure 4 as follows.

 To evaluate in vitro anti-tumor effects based on the anti-PD-L1 mAb in the 4T1 mouse mammary carcinoma model, we measured the proliferation of CD8+ T cells after anti-CD3/CD28 antibody stimulation. As expected, T cell proliferation was dramatically suppressed in 4T1 coculture and restored in 4T1 coculture treated with anti-PD-L1 antibody in a dose-dependent manner (Fig. 4A), suggesting that 4T1 cells have a suppressive activity in the T cell proliferation dependent of PD-1/PD-L1 binding.

(A) After 24 h of exposure to anti-PD-L1 mAb, the 4T1 cells were cocultured with CFSE-labeled splenocytes at 1:10 ratio for 3 days. The cells were harvested and analyzed by flow cytometry.

4) It is not clear why there is no consistence in the used housekeeping gene as sometimes it is Beta actin, then alpha tubulin then GAPDH then alpha actinin

- We thank the reviewer for carefully reading our manuscript. In fact, we used three types of housekeeping genes, such as a-tubulin, GAPDH, and a-actinin. As indicated in the original gel data submitted in a separate file, a-tubulin, GAPDH, and a-actinin have different molecular weights of 55 kD, 37 kD, and 100 kD, respectively. Therefore, in order to simultaneously detect each target protein on the same gel membrane, it was necessary to select appropriate antibody among three antibodies detecting different molecular weight.

5) There is contraction in relation to PD-L2 expression in the presence of absence of NDRG2 as in Figure 1A it seems no difference while in 1F clearly there is decrease to the same extent as PD-L1 although the initial level of expression is different.  PD-L2 data seems not convincing, irrelevant and preferably removed from the manuscript.

- We agree with the reviewer’s comment. Although RT-PCR result in Fig. 1A did not show any significant difference in PD-L2 mRNA expression according to the NDRG2 expression level, the surface expression of PD-L2 protein was decreased by NDRG2 overexpression. However, we thought that degree of change in the expression level was minimal and not convincing when considering competitive binding of both PD-L1 and PD-L2 molecules to PD-1. Nevertheless, we thought that the experimental results need to be presented in the original form to provide an information about differential regulation. We hope reviewer’s understanding.
